# Effects of Complex Antioxidants Added to Chicken Diet on Growth Performance, Serum Biochemical Indices, Meat Quality, and Antioxidant Capacity

**DOI:** 10.3390/ani14030360

**Published:** 2024-01-23

**Authors:** Xiaochun Chen, Dan Zeng, Xiangyi Zeng, Qiufeng Zeng

**Affiliations:** 1Institute of Animal Science, Chengdu Agricultural College, Chengdu 611130, China; chenxiaochun163@163.com; 2Key Laboratory for Animal Disease-Resistance Nutrition of Ministry of Education, Institute of Animal Nutrition, Ministry of Agriculture and Rural Affairs, Sichuan Agricultural University, Chengdu 611130, China; ZD2128024@163.com (D.Z.); zeng20220902@163.com (X.Z.)

**Keywords:** broilers, TBHQ, tea polyphenols, meat quality, antioxidant capacity

## Abstract

**Simple Summary:**

In poultry feed, the oxidative rancidity of lipids is an important factor that affects the quality of feed and the safety of poultry products. Therefore, it is very important to choose and apply antioxidant substances. Tert-butylhydroquinone (**TBHQ**) is a new and highly effective antioxidant for feeds. However, there is limited information about the effects of TBHQ on animals or poultry in vivo. Moreover, tea polyphenols (**TP**) are a natural antioxidant that can prevent and treat diseases by scavenging free radicals and regulating the activity of different types of oxidases in the body. This study comprehensively evaluated the effects of a complex antioxidant, the main components of which are TBHQ and TP, on growth performance, meat quality, and antioxidant capacity in broilers. The obtained results revealed that supplementing diets with optimal amounts of TBHQ and TP contributed to serum and liver antioxidant capacity, thereby improving the growth performance and meat quality of broilers.

**Abstract:**

This study aimed to evaluate the effects of diets supplemented with various levels of complex antioxidants (**CA**) containing tertiary butylhydroquinone (**TBHQ**) and tea polyphenols (**TP**) on growth performance, meat quality of breast and leg muscles, serum biochemistry, and antioxidant capacity of serum, liver, breast meat, jejunum, and ileum in broilers. A total of 600 one-day-old Arbor Acres male broilers with similar body weights were randomly divided into three groups (10 replicates/group, 20 broilers/replicate). Birds in the three experimental groups were fed a basal diet with CA at 0, 300, and 500 mg/kg. The results showed that supplementing with 300 mg/kg CA significantly increased (*p* < 0.05) 42 d BW and 22–42 d ADG, and markedly decreased (*p* < 0.05) 22–42 d F: G ratio in comparison to the control group. Birds fed a diet with 300 mg/kg CA had a higher (*p* < 0.05) pH of chicken meat at 24 h and 48 h post mortem and lower (*p* < 0.05) yellowness values (b*) of chicken meat at 45 min and 24 h post mortem, along with a lower (*p* < 0.05) cooking loss. Supplementing with 300 mg/kg CA significantly increased (*p* < 0.05) serum and liver T-SOD activity, serum T-AOC level, as well as jejunual GST activity, and significantly decreased (*p* < 0.05) liver MDA content when compared with the control group. These results indicate that diet supplementation with 300 mg/kg CA containing TBHQ and TP could improve growth performance and meat quality by increasing the antioxidant capacity of broilers.

## 1. Introduction

Modern poultry breeds selected through genetic breeding have high genetic potential, productivity, and energy requirements [1]. The high energy concentration of their diets requires high levels of lipids (fats and oils). Lipids (fats and oils) are sensitive to oxidation, and are highly susceptible to oxidative deterioration in the process of feed production, use, and storage [2]. In poultry feed, the oxidative rancidity of lipids (oil or polyunsaturated fatty acids, etc.) significantly affects the quality of feed and the safety of poultry products [3]. 

A large number of studies have found that the addition of antioxidant substances to feed has an important role in improving the oxidative stability of feed, as well as the activity of antioxidant enzymes in the bodies of livestock and poultry, extending the shelf-life of animal products [4,5]. Antioxidants, including synthetic antioxidants (for example phenolic antioxidants) and natural antioxidants (for example tocopherols) [6], block the harmful effects of oxygen production and prevent free radicals from damaging the body by trapping and neutralizing them, The most commonly used synthetic antioxidants in food or in feed are phenolic antioxidants [6,7], such as butylated hydroxyanisole (**BHA**), butylated hydroxytoluene (**BHT**), tertiary butylhydroquinone (**TBHQ**) and propyl gallate (**PG**), which can be used for the protection of dietary fats and oils. 

Antioxidant TBHQ is a highly effective antioxidant for feeds, and can prevent oxidative spoilage caused by air oxidation; protect fats, oils, fat-soluble ingredients and other natural components of feeds; and delay the adverse changes caused by the oxidation of foods [8,9]. Numerous studies have shown that the antioxidant TBHQ is essential as a feed preservative in the production of livestock and poultry feeds and can effectively protect feeds against oxidative deterioration [10,11]. In vivo, TBHQ can regulate the expression of antioxidant protein genes by activating proteins; it is often used in medicine as an activator of the nuclear transcription factor E2-related factor-2 [12,13]. However, antioxidant TBHQ is ingested into the bodies of poultry along with the feed consumed, and its effects on the bodies of poultry have not been adequately studied and evaluated.

In recent years, attention has turned to natural antioxidants due to concerns regarding the safety of synthetic antioxidants. Tea polyphenols (**TP**) are a natural antioxidant that can prevent and treat diseases by scavenging free radicals and regulating the activity of different types of oxidases in the body [14]. After entering the animal body, the mechanism by which TP produces antioxidant effects includes the following processes: the increase in activity of antioxidant enzymes; the inhibition of lipid peroxidation; the scavenging of free radicals [15]; and a reduction in oxidation via chelation of metal ions [16]. Complex antioxidants formulated with both synthetic and natural antioxidants, with TBHQ serving as a highly effective feed antioxidant, and TP serving as an important antioxidant for the bodies of poultry, have aroused the interest of this research team. Therefore, the objective of the present study was to evaluate the effects of diets supplemented with an antioxidant complex containing TP and TBHQ on growth performance, meat quality, including regular quality and the chemical composition of breast and leg muscles, serum biochemistry, and the antioxidant capacity of serum, liver, breast meat, jejunum, and ileum, in broilers. 

## 2. Materials and Methods

The Institutional Animal Care and Use Committee (IACUC) of Sichuan Agricultural University approved all procedures used in the study.

### 2.1. Birds, Experimental Design, Diet, and Management

A total of 600 one-day-old male Arbor Acres broiler chickens with a similar initial body weight were randomly distributed into 3 experimental groups with 10 replicates of 20 birds each using a completely randomized design. The birds were fed a basal diet supplemented with 0, 300, and 500 mg/kg complex antioxidants (**CA**), respectively. The trial lasted 42 d, divided into starter (1–21 d) and grower (22–42 d) phases. The main components of the complex antioxidants were ~40%TBHQ, ~20% TP, and ~20% sodium citrate (**SC**), and were provided by MuDan Biochech (Xiamen, China). The composition and nutrient levels of the basal diet are shown in Table 1. The diets were offered in pelleted form. The experimental room temperature and light regimen were maintained at 32 °C to 34 °C and a “23 h on to 1 h off” routine for the first 3 d, gradually decreased to 22 °C at the rate of 2 °C to 3 °C per week, or under “20 h on to 4 h off” for the remainder of the feeding period, respectively. All broilers were housed in cages (2 m length × 0.8 m width, 20 birds/cage) and had free access to water and feed. 

### 2.2. Data and Sample Collection

At 21 and 42 days of age, after 12 h of feed withdrawal, birds were weighed, and the feed intake was obtained by cage. Body weight (**BW**), average daily gain (**ADG**), average daily feed intake (**ADFI**), and feed-to-gain ratio (**F:G**) were calculated for the periods of 1–21 d, 22–42 d, and 1–42 d of age.

On d 42, one bird from each replicate (*n* = 10) with body weights close to the replicate mean were selected and sampled. The blood samples were collected via jugular vein puncturing and placed into tubes without heparin sodium (an anticoagulant). Then, serum was isolated and stored at −20 °C for serum biochemical determination. The birds from which blood was collected were euthanized through exsanguination. Then, liver, pectoral muscle, jejunum, and ileum were collected and stored at −20 °C for antioxidative capacity assays. Some pectorals and leg muscles were then excised and subsequent meat quality parameters were determined, including the pH value, meat color, drip loss, and cooking loss at 45 min, 24 h, and 48 h. Other pectorals and leg muscles were excised and stored at −20 °C for nutrient composition determination. 

### 2.3. Serum Biochemical Indices

The activities of serum alanine aminotransferase (**ALT**) and aspartate aminotransferase (**AST**) as well as the content of serum albumin (**ALB**), total protein (**TP**), high-density lipoprotein cholesterol (**HDL-C**), low-density lipoprotein cholesterol (**LDL-C**), total cholesterol (**TC**), triglyceride (**TG**), and uric acid (**UA**), were analyzed using an automatic biochemical analyzer (HATICHI 7180, Tokyo, Japan).

### 2.4. Determination of Meat Quality

The meat color and pH of the pectoral and leg muscles were determined at 45 min, 24 h and 48 h post mortem, using the method described by [17,18,19]. Briefly, the pH and color parameters, including lightness (L*), redness (a*), and yellowness (b*), were measured at the thickest part of pectoral and leg muscles at 45 min post mortem, and after 24 and 48 h of placement at 4 °C, as previously described. Drip loss was estimated by suspension method: the pectoral and leg muscles were taken and cut into cube-sized (2 cm^3^) meat samples; the meat samples were suspended in self-sealing bags with wire and stored at 4 °C for 24 h. The difference between the initial and final weight of the meat sample as a percentage of the initial weight is the drip loss, according to [20]. Cooking loss was determined by the method described by [21].

### 2.5. Antioxidant Capacity Assays

Tissue samples from liver, pectoral muscle, jejunum, and ileum were prepared as 10% tissue homogenates (1:9), according to the experimental instructions and stored at −20 °C for the determination of antioxidant indices. Total antioxidant capacity (T-AOC, A015-1-2), malondialdehyde (MDA, A003-1-1), total superoxide dismutase (T-SOD, A001-1-1), catalase (enzyme) (CAT, A007-1-1), glutathione-s-transferase (GST, A004-1-1), and hydrogen peroxide (H_2_O_2_) activities were measured in serum and tissue using an enzyme labeling instrument (Tecan Infinite 200 Pro, Männedorf, Switzerland) according to [22]. These kits were purchased from Nanjing Jian Cheng Bioengineering Institute, and all operations were performed according to the kit instructions. 

### 2.6. Statistical Analysis

The data were analyzed by one-way ANOVA using SPSS software (Version 26.0 for windows; SPSS Inc., Chicago, IL, USA). Each replicate was considered an experimental unit. Differences among treatments were detected using Duncan’s multiple range tests. Probability values *≤* 0.05 were considered significant, while values of 0.05 < *p* < 0.1 were considered a tendency. The data are expressed as the mean ± standard deviation (SD).

## 3. Results

### 3.1. Growth Performance

Table 2 shows that feeding with diets supplemented with different dosages of CA have no significant effects (*p* > 0.05) on the growth performance of broilers at 1–21 d of age. However, supplementing with 300 mg/kg CA to diet significantly increased (*p* < 0.05) the BW at 42 d and the ADG from 22 to 42 d or from 1 to 42 d and markedly decreased (*p* < 0.05) the F:G from 22 to 42 d of when compared to that of the control diet.

### 3.2. Serum Biochemical Indices

Table 3 demonstrates that the serum AST activity and UA content were significantly increased (*p* < 0.05) in the 500 mg/kg CA group as compared to the control group. Moreover, serum TC contents were significantly (*p* < 0.05) decreased by dietary supplementation with 300 mg/kg of CA as compared to the control group. Meanwhile, no difference was seen in the contents of TP, ALB, HDL-C, LDL-C, TG, and the activities of ALT and AST in serum (*p* > 0.05) among the three treatments. 

### 3.3. Meat Quality

Table 4 shows the effect of diet supplemented with CA on the breast- and leg-muscle-quality regular parameters of broiler chickens at 42 d of age. The lightness value (L*) and redness (a*) of the meat color of the breast meat at 45 min, 24 h, and 48 h post-mortem were not significantly different *(p* > 0.05) among the three treatment groups. However, compared with broilers fed the control diet and 500 mg/kg CA diet, broilers fed the diet supplemented with 300 mg/kg CA had higher (*p* < 0.05) pH_24h_ and pH_48h_ in breast meat and lower (*p* < 0.05) b*_45min_ and b*_24h_ in breast meat, along with a lower (*p* < 0.05) cooking loss. 

As for leg meat (Table 4), 300 mg/kg CA also had lower (*p* < 0.05) a*_24h_, b*_24h_, and b*_48h_ and a lower (*p* < 0.05) drip loss when compared with the control group. As compared with 500 mg/kg CA group, 300 mg/kg CA significantly increased (*p* < 0.05) the L*_48h_ and markedly decreased (*p* < 0.05) the a*_24h_, a*_48h,_ b*_24h_, and b*_48h_.

As shown in Table 5, the supplemented CA only significantly increased (*p* < 0.05) the content of moisture in breast meat and had no effect *(p* > 0.05) on the content of crude protein and crude fat in chicken meat, and on the content of moisture in leg meat, compared to the control group.

### 3.4. Antioxidant Capacity

Table 6 shows the effect of diet supplemented with CA on the antioxidant capacity in serum, liver, breast meat, jejunum, and ileum of broiler chickens at 42 d of age. Supplementing with 300 mg/kg CA significantly increased (*p* < 0.05) serum and liver T-SOD activity, serum T-AOC concentration as well as jejunual GST activity, and significantly decreased (*p* < 0.05) liver MDA content when compared with the control group. Moreover, supplementing with 500 mg/kg CA significantly increased (*p* < 0.05) liver and breast meat T-SOD activity and significantly decreased (*p* < 0.05) liver and ileal MDA content when compared with the control group. In addition, birds fed a diet supplemented with 300 mg/kg CA presented a higher (*p* < 0.05) serum T-SOD activity, jejunal T-SOD and GST activity and a lower (*p* < 0.05) serum ROM content than birds fed a diet with 500 mg/kg CA.

## 4. Discussion

The present study comprehensively evaluated the effectiveness of CA containing TBHQ and TP on broilers based on growth performance, serum biochemical parameters, and meat quality, including the regular quality and chemical composition of breast and leg muscle, and the antioxidant capacity of serum, liver, breast meat, jejunum, and ileum. We found that broilers fed a diet only supplemented 300 mg/kg CA presented a better growth performance, meat quality, and body antioxidant ability than broilers fed the control diet. The results suggest that the CA containing TBHQ and TP is a good antioxidant product for broilers; however, there is limited information about the effect of TBHQ on growth performance and meat quality in animals or poultry. TBHQ is the active metabolite produced by BHA through a demethylation reaction [23]. In vivo studies have shown that BHA is mainly metabolized to TBHQ in humans, dogs, rats and mouse, and this process is mediated by cytochrome P450 [24]. Imbabi et al. [25] found that supplementing growing rabbits with 50–100 mg/kg BHA may promote growth performance and meat quality via maintaining the redox balance. 

In addition, TP are natural antioxidants typical in flavonoids. Studies have shown that broilers fed diets containing 0.5% to 2% tea polyphenols can improve growth performance, muscle antioxidant capacity and meat quality, and the effect is better than diets supplemented with 0.1% oxytetracycline calcium [26]. Jelveh et al. [27] showed that feeding green tea extract and green tea phytosomes compensated the negative effects of *Eimeria* infection on growth performance in broilers. Chen et al. [28] observed that supplementing 1% green tea powder to the diet could improve the quality of breast meat. Water-holding capacity (WHC) is considered as a key meat quality characteristic of chickens, which is mainly expressed through dripping loss and cooking loss [29]. Consistent with previous findings, we observed that the addition of 300 mg/kg CA to the diet led to a reduction in both dripping loss and cooking loss of the muscle and resulted in a decrease in post-mortem yellowness of the muscle, along with an increase in the postmortem pH value. The postmortem pH value of the muscle increased, indicating improved quality. This may be due to the strong antioxidant activity of the CA. Indeed, Xi et al. [10] found that TBHQ played an important role in attenuating PM2.5-induced pyroptosis and necroptosis in chicken primary alveolar epithelial cells by inhibiting the generation of reactive oxygen species (**ROS**). Liu et al. [30] also found that TBHQ attenuates podocyte injury in diabetic nephropathy by inhibiting NADPH oxidase-derived ROS generation via the nuclear factor E2-related factor-2 /hemeoxygenase-1 signaling pathway. Studies have shown that TBHQ can disrupt the metabolic processes of exogenous toxic compounds and reduce the synthesis of toxic metabolites in the body, thereby reducing oxidative stress damage [9]. Yan et al. [14] conducted a review about antioxidant mechanism of TP and its impact on health benefits. In vivo experiments shoed that TP can increase levels of rat serum catalase, glutathione peroxidase and SOD, and can reduce the production of MDA as well as the content of TC in serum. Consistent with the previous studies, the results of our study demonstrated that dietary supplementation of CA increased the activities of antioxidant enzymes in the serum, liver and jejunum of broilers and decreased the liver MDA level as well as serum TC content. Therefore, we suggest that the CA containing TBHQ and TP improved the growth performance and meat quality by increasing the antioxidant capacity of broilers.

It is worth noting that broilers fed a diet with 500 mg/kg CA did not present further improvement in growth performance and meat quality in comparison to diets with 300 mg/kg CA. Meanwhile, our study observed a significant increase in serum AST and UA levels in the 500 mg/kg dose group compared to the control group. Serum biochemical indicators are frequently used to reflect the changes in material metabolism and tissues and organs function in broilers [31]. When the body’s liver is damaged or has metabolic disorders, ALT and AST enter the blood circulation from liver cells. Therefore, a high AST level in the serum could be an important indicator of liver damage [32]. These findings suggest that the high dose of CA may have a detrimental effect on the liver of broiler chickens on d 42. Furthermore, AST’s involvement in protein or amino acid metabolism and the urea cycle could explain the high uric acid levels observed in 42 d birds with the same 500 mg/kg CA dose [33]. Synthetic phenolic antioxidants are widely used in feed; considering their potential risks, it is very important to monitor the content of these in feed. The European Food Safety Authority [34] established that BHA and TBHQ, individually or in combination, in the same foods, can be used at a maximum permitted level of 200 mg/kg. Limits to the quantity of BHT, BHA and TBHQ, individually, or in combination in the same feed, are all 150 mg/kg in feed according to Chinese Feed Hygiene Standards. However, TBHQ alone, or in combination only with BHT and/or BHA, can be used with a maximum 200 mg/kg by weight of the fat and oil content and not by the weight of the commercial complete feed, according to the Code of Federal Regulations [35]. Therefore, attention should be paid to the appropriate supplemented amount of TBHQ in broiler commercial feed.

## 5. Conclusions

In conclusion, diet supplementation with 300 mg/kg CA containing TBHQ and TP improved the growth performance and meat quality by increasing the antioxidant capacity of broilers. However, when using CA containing TBHQ and TP in broiler feed, attention may need to be paid to the appropriate amount of the additions in vitro and in vivo. It is possible to prepare a complex antioxidant formulation to increase antioxidant capacity by using synthetic or natural antioxidant substances.

## Figures and Tables

**Table 1 animals-14-00360-t001:** The composition and nutrient levels of the basal diets (air dry basis).

Ingredients	Content, %	
	1 to 21 d	22 to 42 d
Corn	49.56	51.40
Soybean meal	38.00	35.32
Wheat flours	1.20	4.20
Poultry fat	4.00	4.80
Corn gluten meal	1.65	0.00
Calcium carbonate	1.26	1.33
Dicalcium phosphate	1.90	1.32
DL-Methionine	0.18	0.21
L-Lysine.-HCL	0.00	0.09
L-Threonine	0.00	0.04
Vitamin premix 1	0.03	0.03
Mineral premix 2	0.20	0.20
Choline chloride (50%)	0.15	0.15
Sodium chloride	0.30	0.30
Rice bran	1.57	0.61
Total	100.00	100.00
Calculated nutrient levels, %		
ME, MJ/kg	12.54	12.97
Crude protein	21.50	20.00
Calcium	1.00	0.90
Non-phytate phosphorus	0.45	0.35
Digestible lysine	1.15	1.00
Digestible methionine	0.50	0.40
Digestible threonine	0.83	0.72

^1^ Vitamin premix provides the following per kg of final diet: vitamin A, 8000 IU; vitamin D_3_, 2000 IU; vitamin E, 20 IU; vitamin K_3_, 0.5 mg; vitamin B_1_, 2.0 mg; vitamin B_2_, 8.0 mg; vitamin B_6_, 3.5 mg; vitamin B_12_, 0.01 mg; pantothenic acid, 10.0 mg; niacin, 35.0 mg; folic acid, 0.55 mg; biotin, 0.18 mg. ^2^ Mineral premix provides the following per kg of final diet: copper (CuSO_4_·5H_2_O) 8.0 mg; iron (FeSO_4_·H_2_O) 80.00 mg; zinc (ZnSO_4_·H_2_O) 80.00 mg; manganese (MnSO_4_·H_2_O) 100.00 mg; iodine (KI) 0.70 mg; selenium (Na_2_SeO_3_) 0.30 mg.

**Table 2 animals-14-00360-t002:** Effect of diets supplemented with complex antioxidants on growth performance of broiler (*n* = 10) ^1^.

Items	Dietary Complex Antioxidant Levels, mg/kg			*p*-Values
	0	300	500	
Body weight (BW), g/bird				
1 d	43.50 ± 0.48	43.65 ± 0.56	43.60 ± 0.40	0.788
21 d	904.8 ± 27.69	922.6 ± 22.88	892.8 ± 29.86	0.121
42 d	2903 ± 84.64 ^b^	3022 ± 85.30 ^a^	2940 ± 82.07 ^ab^	0.028
Average daily gain (ADG), g/bird/d				
1–21 d	41.02 ± 1.33	41.45 ± 1.23	40.44 ± 1.42	0.294
22–42 d	91.36 ± 4.27 ^b^	96.13 ± 2.53 ^a^	92.08 ± 4.13 ^b^	0.037
1–42 d	68.08 ± 2.02 ^b^	71.07 ± 1.60 ^a^	69.35 ± 2.19 ^ab^	0.020
Average daily feed intake (ADFI), g/bird/d				
1–21 d	58.27 ± 1.46	58.84 ± 1.64	57.53 ± 1.80	0.257
22–42 d	154.56 ± 4.69	155.78 ± 6.26	154.34 ± 5.34	0.882
1–42 d	108.31 ± 1.59	110.96 ± 3.84	110.20 ± 1.91	0.158
Feed to gain ratio (F:G), g/g				
1–21 d	1.42 ± 0.02	1.42 ± 0.02	1.42 ± 0.02	0.655
22–42 d	1.69 ± 0.06 ^a^	1.62 ± 0.05 ^b^	1.67 ± 0.05 ^ab^	0.045
1–42 d	1.59 ± 0.03	1.56 ± 0.04	1.59 ± 0.03	0.177

^a,b^ Means within a row with different superscripts are different at *p* < 0.05. ^1^ Values are the mean of 10 replicates of 20 chickens each.

**Table 3 animals-14-00360-t003:** Effect of diets supplemented with complex antioxidants on serum biochemical parameters of broiler chickens at 42 d of age (*n* = 10) ^1^.

Items	Dietary Complex Antioxidant Levels, mg/kg			*p*-Values
	0	300	500	
TP ^2^ g/L	23.18 ± 4.48	23.50 ± 5.50	25.09 ± 6.26	0.757
ALB g/L	9.45 ± 1.66	8.90 ± 1.64	9.59 ± 1.70	0.658
UA umol/L	122.09 ± 16.77 ^b^	157.54 ± 63.93 ^ab^	203.54 ± 72.98 ^a^	0.022
ALT U/L	4.62 ± 1.56	4.35 ± 0.85	5.31 ± 1.35	0.281
AST U/L	230.04 ± 49.59 ^b^	224.97 ± 51.02 ^b^	329.05 ± 107.56 ^a^	0.038
HDL-C mmol/L	2.66 ± 0.35	2.51 ± 0.46	2.81 ± 0.42	0.294
LDL-C mmol/L	0.66 ± 0.19	0.71 ± 0.23	0.68 ± 0.24	0.885
TC mmol/L	4.46 ± 0.60 ^a^	3.79 ± 0.49 ^b^	3.98 ± 0.42 ^ab^	0.043
TG mmol/L	0.24 ± 0.09	0.24 ± 0.05	0.28 ± 0.08	0.483

^a,b^ Means within a row with different superscripts are different at *p* < 0.05. ^1^ Values are the mean of 10 replicates of one chicken each. ^2^ Abbreviations: TP, total protein; ALB, albumin; UA, uric acid; ALT, alanine aminotransferase; AST, aspartate aminotransferase; HDL-C, high-density lipoprotein; LDL-C, low-density lipoprotein; TC, total cholesterol; TG, triglyceride.

**Table 4 animals-14-00360-t004:** Effects of diets supplemented with complex antioxidants on breast- and leg-meat-quality parameters of broiler chickens at 42 d of age (*n* = 10) ^1^.

Items	Dietary Complex Antioxidant Levels, mg/kg			*p*-Values
	0	300	500	
Chicken meat				
L*_45min_	66.26 ± 2.80	65.51 ± 3.83	65.48 ± 4.75	0.701
L*_24h_	51.03 ± 2.69	51.64 ± 2.56	50.92 ± 2.29	0.509
L*_48h_	52.96 ± 2.48	51.99 ± 2.57	52.31 ± 2.45	0.391
a*_45min_	7.57 ± 2.27	7.17 ± 1.72	7.53 ± 2.10	0.733
a*_24h_	6.87 ± 1.58	6.84 ± 1.64	7.13 ± 1.48	0.745
a*_48h_	7.61 ± 1.60	7.93 ± 1.59	7.10 ± 1.49	0.144
b*_45min_	11.26 ± 1.93 ^a^	9.88 ± 1.71 ^b^	10.61 ± 2.34 ^ab^	0.043
b*_24h_	9.32 ± 0.97 ^a^	8.70 ± 1.04 ^b^	9.35 ± 1.16 ^a^	0.048
b*_48h_	9.29 ± 1.10	9.67 ± 1.21	9.48 ± 1.32	0.506
pH_45min_	5.98 ± 0.14 ^ab^	6.04 ± 0.17 ^a^	5.91 ± 0.19 ^b^	0.017
pH_24h_	5.80 ± 0.10 ^b^	5.86 ± 0.09 ^a^	5.77 ± 0.11 ^b^	0.013
pH_48h_	5.76 ± 0.09 ^b^	5.83 ± 0.09 ^a^	5.77 ± 0.12 ^b^	0.046
Drip loss_24h_/%	4.88 ± 2.05	3.82 ± 1.10	3.43 ± 1.29	0.196
Cooking loss/%	23.47 ± 1.79 ^ab^	21.76 ± 1.95 ^b^	25.26 ± 3.25 ^a^	0.030
Leg meat				
L*_45min_	64.40 ± 3.688	64.28 ± 4.86	63.57 ± 4.02	0.739
L*_24h_	52.94 ± 3.16	52.35 ± 2.77	51.94 ± 4.08	0.567
L*_48h_	53.41 ± 3.40 ^ab^	54.44 ± 2.61 ^a^	51.98 ± 4.44 ^b^	0.045
a*_45min_	7.20 ± 1.92	7.76 ± 1.94	8.38 ± 1.64	0.780
a*_24h_	13.81 ± 2.41 ^a^	11.86 ± 1.83 ^b^	14.13 ± 2.13 ^a^	<0.01
a*_48h_	13.36 ± 2.24 ^b^	12.32 ± 1.63 ^b^	14.91 ± 3.18 ^a^	<0.01
b*_45min_	9.15 ± 2.35 ^a^	7.85 ± 2.57 ^ab^	7.05 ± 2.88 ^b^	0.020
b*_24h_	10.29 ± 1.87 ^a^	9.23 ± 1.75 ^b^	10.45 ± 1.79 ^a^	0.028
b*_48h_	11.41 ± 1.69 ^a^	9.95 ± 1.45 ^b^	10.97 ± 1.61 ^a^	<0.01
pH_45min_	6.24 ± 0.10	6.27 ± 0.10	6.22 ± 0.11	0.259
pH_24h_	6.26 ± 0.16	6.29 ± 0.10	6.30 ± 0.17	0.699
pH_48h_	6.25 ± 0.13	6.27 ± 0.09	6.27 ± 0.12	0.875
Drip loss_24h_/%	6.06 ± 1.82 ^a^	3.79 ± 1.86 ^b^	3.69 ± 1.68 ^b^	0.024
Cooking loss/%	20.54 ± 4.03	17.86 ± 4.97	19.78 ± 1.43	0.345

^a,b^ Means within a row with different superscripts are different at *p* < 0.05. ^1^ Values are the mean of 10 replicates of one chicken each.

**Table 5 animals-14-00360-t005:** Effect of diets supplemented with complex antioxidants on the nutrient composition of breast and leg muscles of broiler chickens at 42 d of age (*n* = 10) ^1^.

Item	Dietary Complex Antioxidant Levels, mg/kg			*p*-Values
	0	300	500	
Chicken meat,%				
Moisture	74.25 ± 0.88 ^b^	75.39 ± 1.11 ^a^	75.21 ± 1.05 ^a^	0.043
Crude protein	23.03 ± 0.85	22.08 ± 1.00	22.33 ± 1.08	0.125
Crude fat	1.81 ± 0.41	1.77 ± 0.31	1.99 ± 0.27	0.362
Leg meat,%				
Moisture	77.01 ± 0.61	77.11 ± 0.98	76.94 ± 0.82	0.902
Crude protein	18.85 ± 0.66	18.85 ± 0.55	19.50 ± 0.50	0.054
Crude fat	2.29 ± 0.57	1.90 ± 0.55	1.78 ± 0.48	0.155

^a,b^ Means within a row with different superscripts are different at *p* < 0.05. ^1^ Values are the mean of 10 replicates of one chicken each.

**Table 6 animals-14-00360-t006:** Effect of diets supplemented with complex antioxidants on antioxidant capacity of broiler chickens at 42 d of age (*n* = 10) ^1^.

Item	Dietary Complex Antioxidant Levels, mg/kg			*p*-Values
	0	300	500	
Serum				
CAT ^2^, U/mL	4.33 ± 1.91	4.83 ± 1.30	4.31 ± 3.68	0.794
T-SOD, U/mL	354.84 ± 13.86 ^b^	468.87 ± 45.43 ^a^	355.82 ± 30.61 ^b^	<0.01
T-AOC, mM	0.603 ± 0.11 ^b^	0.718 ± 0.09 ^a^	0.630 ± 0.10 ^ab^	0.047
GST, U/mL	22.32 ± 3.54	22.16 ± 1.96	21.86 ± 3.60	0.956
ROM, mmolH_2_O_2_/L	38.39 ± 9.55 ^ab^	33.61 ± 9.03 ^b^	46.92 ± 12.17 ^a^	0.041
MDA, nmol/mL	3.22 ± 1.46	2.65 ± 1.01	3.10 ± 0.87	0.698
Liver				
CAT, U/mgprot	29.50 ± 9.28	30.76 ± 5.88	22.28 ± 5.84	0.063
T-SOD, U/mgprot	499.28 ± 56.61 ^b^	709.98 ± 98.80 ^a^	641.82 ± 102.33 ^a^	<0.01
GST, U/mgprot	111.04 ± 31.52	123.78 ± 25.24	128.60 ± 13.46	0.348
MDA, nmol/mlprot	0.82 ± 0.28 ^a^	0.49 ± 0.23 ^b^	0.42 ± 0.19 ^b^	<0.01
Chicken meat				
CAT, U/mgprot	12.12 ± 7.78	14.76 ± 11.84	20.92 ± 14.49	0.368
T-SOD, U/mgprot	51.22 ± 7.15 ^b^	60.69 ± 12.00 ^ab^	66.69 ± 14.80 ^a^	0.031
MDA, nmol/mlprot	0.60 ± 0.15	0.60 ± 0.21	0.52 ± 0.18	0.574
Jejunum				
T-SOD, U/mgprot	137.35 ± 21.87 ^ab^	170.60 ± 30.97 ^a^	131.52 ± 35.07 ^b^	0.046
GST, U/mgprot	185.47 ± 47.67 ^b^	239.25 ± 53.22 ^a^	184.64 ± 44.09 ^b^	0.038
MDA, nmol/mlprot	0.58 ± 0.21	0.43 ± 0.31	0.52 ± 0.21	0.437
Ileum				
T-SOD, U/mgprot	84.37 ± 13.43	80.17 ± 15.04	90.21 ± 20.29	0.526
GST, U/mgprot	161.24 ± 35.41	171.43 ± 39.14	172.66 ± 36.07	0.772
MDA, nmol/mlprot	0.45 ± 0.16 ^a^	0.37 ± 0.12 ^ab^	0.28 ± 0.10 ^b^	0.033

^a,b^ Means within a row with different superscripts are different at *p* < 0.05. ^1^ Values are the mean of 10 replicates of one chicken each. ^2^ Abbreviations: CAT, catalase; T-SOD, total superoxide dismutase; T-AOC, total antioxidant capacity; ROM, hydrogen peroxide per liter of serum; GST, glutathione S-transferase; MDA, malondialdehyde.

## Data Availability

Data are contained within the article.

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
