# Peer review of "Effects of Complex Antioxidants Added to Chicken Diet on Growth Performance, Serum Biochemical Indices, Meat Quality, and Antioxidant Capacity"

_animals, 2024, doi:10.3390/ani14030360_

Round 1
Reviewer 1 Report
Comments and Suggestions for Authors
The manuscript is very interesting, it was well written and is easy to follow.
Concerning my questions:
1) As far as I understood, 2 treatments were done, with 0, 300 and 500 ppm of a complex antioxidant composed of 40% TBHQ, 20% natural antioxidant (tea polyphenol-TP), and 20% sodium citrate. I was wondering why haven´t you tested a group only with natural antioxidant? As you mentioned is a tendency.
2) If you report that European Food Safety Authority, (2016) in lines 375-377 established that BHA and TBHQ, individually or combined can be used at a maximum permitted level of 200 mg/kg, why did you tested concentrations of 300 and 500 mg/kg? What is the cost-benefit of adding this amount of antioxidant in a feed and how much would it cost to the producer/farmer?
3) Please add another legislations, such as the American one (https://www.govinfo.gov/content/pkg/CFR-2023-title9-vol2/pdf/CFR-2023-title9-vol2.pdf), which establishes TBHQ (tertiary butylhydroquinone), alone or in combination (only with BHT and/or BHA), with a maximum 0.02 % by weight of the fat and oil content. If you wanna see the General collection is here: https://www.govinfo.gov/app/collection/cfr/2023/title9 regarding Animals and Animal Products.
4) Replace the term "breast meat" by "chicken meat" or "chicken muscle", because it is widely used by poultry farming and Associations of Animal Protein worldwide.
5) The terms “broiler chicken” is an old, and now redundant, industry term, and for the benefit of consumers and industry, it should be changed in the title. Along the manuscript, you can change the terms too. Or use broilers or use chickens, both together are redundant.
I would suggest a title such as:
"Effects of complex antioxidants added to chicken diet on Growth Performance, Serum Biochemical Indices, Meat Quality, and Antioxidant Capacity".
6) Why type of flour was used in the diet (Table 1, third line)? I did not find any other mention along the text about it.

Author Response
Dear reviewer,
We do really appreciate the suggestions or comments from you. Thank you for your hard work! Happy New Year!
We have revised the manuscript one by one according to your suggestions or have given further explanation/justification for subsequent edits. We used yellow to mark the revised places, please check the revised manuscript. The answer of suggestions one by one are the attach.
Best regards,
Qiufeng Zeng & Xiaochun Chen
14/01/2024

Reviewer 2 Report
Comments and Suggestions for Authors
Dear Authors,
Abstract
Lines 52-53: Replace “growth performance” with “body weight”
Materials and Methods
Lines 122-125: Were the broilers reared on the floor or in a cage?
Lines 121-137: Please specify stocking density.
Line 128-129: “40%TBHQ, 20% TP, and 20% sodium”. What standards did you use for determining these rates?
Lines 212: If you are giving the standard deviation, you must also give the (n) numbers. Or you can just give the standard error.
Results
Lines 218-221: This result is not understandable, please write more clearly.
Table 3: Other characteristics whose differences are insignificant between groups in the table should also be written.
Lines 265-267: Either delete this sentence or indicate the difference between the groups in the Table 5.
Table 5: Other characteristics whose differences are insignificant between groups in the table should also be written.
Discussions
Line 303: It is seen that the performance characteristics only affect the body weight on the 42nd day, so it is not appropriate to make this generalization. It is more appropriate to write body weight instead of growth performance.
Line 354: Replace “Therefor” with “Therefore”
Line 355: “replace “growth performance” with “body weight”
The treatments appear to have a reducing effect on total cholesterol. Please add discussion about this.
Author Response
Dear Reviewer :
We do really appreciate the suggestions or comments from you. Thank you for your hard work! Happy New Year!
We have revised the manuscript one by one according to your suggestions or have given further explanation/justification for subsequent edits. We used yellow to mark the revised places, please check the revised manuscript. The answer of suggestions one by one are the attach.
Best regards,
Qiufeng Zeng & Xiaochun Chen
14/01/2024

Reviewer 3 Report
Comments and Suggestions for Authors
The paper is well written, the experiment is well thought out, and the results are clearly presented. The subject is interesting and relevant, having in mind that addition of feed additives is necessary in modern poultry production. Objectives are clearly defined, and the experimental design is appropriate.
However, I think that some parts of the paper should be improved by adding some more facts about the potential detrimental effects of TBHQ which are pointed out in the research of other authors. Also, almost all attention is focused on the 300 mg group, with very little comment on the 500 mg group.
Specific comments:
Abstract (general): you didn’t mention anything about the results of 500 mg/kg CA group. Please add something about the results of that group in Abstract
Line 17: I would rather say “cause” or “factor” than the “reason”
Line 34 (and in all paper): what is regular quality of the meat? This is not defined, so please specify the traits which you measured to evaluate meat quality.
Line 43: F:G ratio
Line 52: I am not sure why you stated that …300 mg/kg CA could improve growth performance, when your results indicated that 300 mg/kg CA improved growth performance and meat quality,…
Introduction
Line 76: You said “Recently”… but the paper you refer to is from 2012. Also, this paper is cited to support the statements about the effect of some antioxidants in feed and food, but the paper is about the chlorination of water. Please add some more literature about the subject.
Line 86: You said “Numerous studies”… but you cited only one (No. 10). Please add more literature.
In the chapter (Lines 82 to 95) you paid no attention to the potential detrimental effects of using TBHQ as additive, even though papers that you cited in this chapter refer mostly to its toxicity. It would be very useful for the readers if you could also explain the part about its potential toxicity.
Table 2: It should be indicated what the values after +/- mean. Is it SD? I think that results of body weight at 21 and 42 days should be presented without decimal places.
Discussion: almost all attention is focused on the 300 mg group, with very little comment on the 500 mg group. I think you should add some more comments on 500 mg group.
Line 329: For example, you stated that “addition of 300 mg/kg CA led to a reduction in dripping loss and cooking loss…” but the drip loss was the smallest in 500 mg group. You should mention that in the discussion.
Conclusions: You should mention the effect of 500 mg/CA which was expressed in increased ALT and UA, and to emphasize the importance of appropriate dosage of CA.
Literature: No. 29 is the same as 11.
Author Response
Dear Reviewer :
We do really appreciate the suggestions or comments from you. Thank you for your hard work!
We have revised the manuscript one by one according to your suggestions or have given further explanation/justification for subsequent edits. We used yellow to mark the revised places, please check the revised manuscript. The answer of suggestions one by one are the attach.
Best regards,
Qiufeng Zeng & Xiaochun Chen
14/01/2024
